# Catching Two Birds with One Stone: Reward Shaping with Dual Random Networks for Balancing Exploration and Exploitation

Haozhe Ma [1]  Fangling Li [2]  Jing Yu Lim [1]  Zhengding Luo[*][3]  Thanh Vinh Vo [1]  Tze-Yun Leong [1]

## Abstract

Existing reward shaping techniques for sparse-reward reinforcement learning generally fall into two categories: novelty-based exploration bonuses and significance-based hidden state values. The former promotes exploration but can lead to distraction from task objectives, while the latter facilitates stable convergence but often lacks sufficient early exploration. To address these limitations, we propose Dual Random Networks Distillation (DuRND), a novel reward shaping framework that efficiently balances exploration and exploitation in a unified mechanism. DuRND leverages two lightweight random network modules to simultaneously compute two complementary rewards: a novelty reward to encourage directed exploration and a contribution reward to assess progress toward task completion. With low computational overhead, DuRND excels in high-dimensional environments with challenging sparse rewards, such as Atari, VizDoom, and MiniWorld, outperforming several benchmarks.

## 1. Introduction

Reinforcement Learning (RL) involves an agent optimizing policies to maximize cumulative rewards within an environment, without any model of its dynamics (Sutton & Barto, 2018). One pivotal challenge in RL is balancing exploration and exploitation, both are critical stages for effective learning. Sufficient exploration is vital, particularly in tasks with extremely sparse rewards where feedback is only available at the end of each episode. In such scenarios, directed exploration is necessary for agents to identify diverse samples that

potentially yield positive outcomes (Ladosz et al., 2022). Conversely, in later training phases, exploitation becomes crucial to reinforce behaviors that are known to be successful in maximizing rewards, ensuring stable convergence. It is important to leverage information that aligns closely with the agent's original goals.

One well-studied line of work is reward shaping (RS), which designs additional rewards to supplement the sparse environmental rewards, providing fine-grained, immediate feedback (Sorg et al., 2010a;b; Ibrahim et al., 2024; Lidayan et al., 2025). Introducing *exploration bonus* as auxiliary rewards stands out as a promising RS approach. By assigning higher rewards to novel states, it explicitly encourages agents to explore under-explored regions (Baldassarre et al., 2013; Bellemare et al., 2016; Zheng et al., 2018; Devidze et al., 2022). However, since novelty does not necessarily correlate with meaningfulness or align with the agent's ultimate goals, continuously rewarding novelty may cause agents to disproportionately focus on samples from sub-optimal trajectories or even dangerous regions during the stabilization stages, thereby distracting them from convergence to optimal policies. The well-known "noisy TV" problem is an example where agents become captivated by highly novel but irrelevant TV channels in a maze navigation task (Mavor-Parker et al., 2022). This highlights that agents need to gradually recover from novelty rewards and shift towards exploitation as training progresses.

Alternatively, *hidden state value* based RS methods primarily develop task-relevant signals to quantify the extent to which states contribute to achieving higher environmental rewards and their inherent significance, e.g., the distance to the goal state and the priorities of key points, thereby enhancing exploitation (Trott et al., 2019; Memarian et al., 2021; Park et al., 2023; Ma et al., 2024b; 2025). Unlike exploration-centric approaches, they typically rely on their backbone algorithms' exploration strategies. Although highly efficient in exploiting known experiences, they often struggle in environments with extremely sparse rewards due to the lack of directional exploration guidance.

Building on the insights from both exploration bonus and hidden state value approaches, a natural research question arises: *Can we devise a unified mechanism that com-*

[1]School of Computing, National University of Singapore, Singapore [2]Department of Statistics and Data Science, National University of Singapore, Singapore [3]School of Electrical and Electronic Engineering, Nanyang Technological University, Singapore. Correspondence to: Zhengding Luo <luoz0021@e.ntu.edu.sg>.

*Proceedings of the 42$^{nd}$ International Conference on Machine Learning*, Vancouver, Canada. PMLR 267, 2025. Copyright 2025 by the author(s).

*putes both types of rewards efficiently, with minimal computational overhead and design efforts, thereby achieving an effective balance between exploration and exploitation?* To this end, we propose the **Du**al **R**andom **N**etworks **D**istillation (**DuRND**, pronounced "Durian") framework[1], inspired by Random Network Distillation (RND), which is originally developed to quantify the novelty of a state relative to previously encountered ones (Burda et al., 2018). DuRND is designed to enable efficient exploration and stable convergence in sparse-reward RL. It incorporates two separate Random Network (RN) modules: a *positive RN* module for states that are potentially guiding agents toward task completion or contributing significantly to obtaining original rewards with high state values; and a *negative RN* module for states that offer little benefit in achieving rewards or may even mislead agents away from their objectives. States are classified as positive or negative naturally by the sparse environmental rewards. With the dual RN modules, DuRND concurrently derives two types of rewards: (a) a *novelty reward*, which evaluates how distinct a state is from all previously encountered states, and (b) a *contribution reward*, which assesses a state's hidden value of getting higher rewards, tightly aligning with agents' original goals. The main contributions of this paper are summarized as follows:

(i) We propose DuRND, a framework that leverages two RN modules to jointly compute a novelty reward to encourage directed exploration and a contribution reward to enhance experience exploitation. Involving two types of rewards, DuRND achieves exploration-efficient and convergence-stable learning in challenging sparse-reward tasks.

(ii) DuRND operates with minimal computational overhead. Unlike some RS methods that rely on auxiliary agents, large historical state buffers, or pseudo-count estimations (Bellemare et al., 2016; Ostrovski et al., 2017; Mguni et al., 2023; Ma et al., 2024b), DuRND uses only two lightweight RN modules, making it highly scalable in high-dimensional environments.

(iii) The effectiveness and efficiency of DuRND are validated across a variety of sparse-reward tasks with high-dimensional states, demonstrating its superior performance compared to several benchmarks.

## 2. Background

**Reinforcement Learning (RL)** operates within the framework of **Markov Decision Processes (MDP)**, formalizing the interaction between an agent and an environment as a tuple $\langle S, A, T, R, \gamma \rangle$. $S$ and $A$ are state space and action space, respectively, $T : S \times A \times S \to [0, 1]$ is the transition

---

[1]The source code is accessible at: https://github.com/mahaozhe/DuRND

function, $R : S \to \mathbb{R}$ is the reward function, and $\gamma \in [0, 1)$ is the discount factor. This paper studies stochastic policies $\pi : S \times A \to [0, 1]$ that maximize the expected discounted return $\mathbb{E}_\tau[\sum_{t=0}^{\infty} \gamma^t R(s_t)]$, where $\tau = (s_0, a_0, s_1, a_1, \dots)$ is a trajectory of states and actions, and $s_{t+1} \sim T(\cdot|s_t, a_t)$, $a_t \sim \pi(\cdot|s_t)$. Common techniques in RL encompass value-based methods, policy-based methods, and their hybrid, actor-critic methods (Sutton & Barto, 2018).

**Random Network Distillation (RND)** motivates agents to explore less frequently visited states by using novelty as an exploration bonus (Burda et al., 2018). RND introduces two neural networks: a fixed, randomly initialized *target network* $f(s) : S \to \mathbb{R}^k$, and a trainable *predictor network* $\hat{f}(s; \theta) : S \to \mathbb{R}^k$. Both networks map a state $s \in S$ to a $k$-dimensional feature embedding. The predictor network is trained to minimize the mean squared error (MSE) $e(s) = \|\hat{f}(s; \theta) - f(s)\|^2$ through gradient descent. This MSE also quantifies a state's novelty, as higher errors occur for states dissimilar to those the predictor has seen, thereby, the exploration bonus is defined as $R^{\mathrm{rnd}}(s) = e(s)$. As the predictor is trained on samples collected by the agent, it gradually develops a "memory" of the states encountered.

## 3. Related Work

**Exploration Bonuses** as shaped rewards have been widely used to guide the exploration directions. The most intuitive method is the count-based approach, where the exploration bonus is assessed by each state's visitation frequency (Strehl & Littman, 2008). To adapt state counting to continuous or unlimited state spaces, pseudo-counts were introduced (Bellemare et al., 2016), with several works studied how to estimate the pseudo-counts (Fox et al., 2018; Badia et al., 2020; Devidze et al., 2022; Yuan et al., 2021; Luo et al., 2024). Specifically, Bellemare et al. (2016) derived the visited counts from the Context Tree Switching model, Fu et al. (2017) used exemplar models for implicit density estimation, Tang et al. (2017) discretized continuous states using hash functions, and Machado et al. (2020) incorporated the successor representation. Although tractable, these methods often require extensive storage resources or inference time. Following the pseudo-count concept, neural network-based methods have been developed. Ostrovski et al. (2017) used PixelCNN (Van den Oord et al., 2016) for density estimation; Martin et al. (2017) used the feature representation from value approximation networks; Lobel et al. (2023) derived the pseudo-counts by averaging samples from the Rademacher distribution; and Burda et al. (2018) introduced Random Network Distillation to assess state novelty, while Yang et al. (2024) further improved the precision of bonus allocation.

**Hidden Values** as shaped rewards effectively guide the optimization direction of agents to accelerate the convergence.

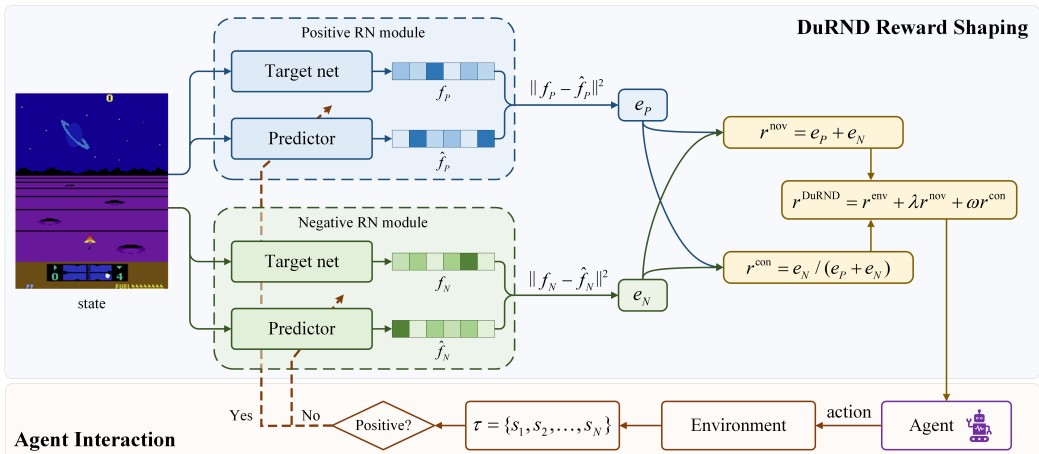

Figure 1: An overview of the Dual Random Networks Distillation (DuRND) framework. The state is processed through both *positive* and *negative* RN modules to derive errors that reflect its frequency under two scenarios. The two errors jointly form both the novelty reward and the contribution reward to support agent training. The RN modules are updated correspondingly using states associated with positive or negative outcomes, as indicated by the sparse environmental rewards.

One common approach is to build reward models from expert demonstrations (inverse reinforcement learning) (Arora & Doshi, 2021; Cheng et al., 2021) or human feedback (RLHF) (Christiano et al., 2017), which have been popularly applied in robotic control (Ellis et al., 2021; Schultheis et al., 2021; Bıyık et al., 2022) and large language models (LLMs) (Sumers et al., 2021; Ghosal et al., 2023; Wu et al., 2023; Hwang et al., 2023; Fang et al., 2023; 2025; Dai et al., 2024). However, these methods require considerable human-generated data, which is often challenging to obtain, especially in highly specialized or advanced domains. Another line of research has emerged to derive beneficial information directly from the agent's own learning experiences (Zheng et al., 2018; Hu et al., 2020; Park et al., 2023; Gupta et al., 2023; Zhong et al., 2024). Representatively, Trott et al. (2019) used the state-goal distance as a heuristic, Memarian et al. (2021) ranked different trajectories via a trained classifier indicated by the preferences, Ma et al. (2024b;a) introduced an assistant reward agent to collaboratively generate rewards guiding the policy agent, Ma et al. (2025) derived the success ratio via a Thompson sampling approach to evaluate a state's contribution to task completion. However, these methods usually rely on the underlying algorithm's exploration strategies, which may lead to suboptimal policies due to insufficient sample diversity. We seek to combine the hidden value and exploration bonus, aiming to achieve efficient exploration and stable convergence.

Other reward shaping methods have been explored, leveraging diverse strategies. Potential-based algorithms defined rewards as the temporal difference of a potential function, ensuring that the optimal policy remains consistent with the original MDP (Asmuth et al., 2008; Devlin & Kudenko, 2012; Koprulu et al., 2024; Adamczyk et al., 2025). Infor-

mation gain based approaches used the prediction errors in dynamics to model how surprising the states are to motivate exploration (Houthooft et al., 2016; Pathak et al., 2017; Hong et al., 2018; Burda et al., 2019; Sun et al., 2022). However, they require an environmental transition model, which makes them challenging in adapting to large-scale scenarios with complex dynamics. Additionally, some studies incorporated concepts of uncertainty or diversity (Eysenbach et al., 2019; Pathak et al., 2019; Raileanu & Rocktäschel, 2020), or involved multiple agents or hierarchical structures to shape rewards (Stadie et al., 2020; Vo et al., 2022a;b; Yi et al., 2022; Fang et al., 2022; Mguni et al., 2023; Ma et al., 2023; 2024c; Zhong et al., 2025a;b).

## 4. Reward Shaping via Dual RN Modules

In the proposed DuRND framework, the shaped reward is defined by integrating two auxiliary rewards:

$$R^{\text{DuRND}}(s) := R^{\text{env}}(s) + \lambda R^{\text{nov}}(s) + \omega R^{\text{con}}(s), \quad (1)$$

where $R^{\text{env}}(s)$ is the environmental reward, $R^{\text{nov}}(s)$ is the *novelty reward*, serving as an exploration bonus, and $R^{\text{con}}(s)$ is the *contribution reward*, capturing a states' hidden value in achieving overall performance. We set $\lambda = \omega = 0.5$ to balance the scales of both auxiliary rewards, following the standard setting of vanilla RND (Burda et al., 2018). Here, Both $R^{\text{nov}}(s)$ and $R^{\text{con}}(s)$ are jointly computed by two separate Random Network (RN) modules, referred to as the *positive RN* and the *negative RN*. They are updated using states that yield favorable and unfavorable outcomes, respectively, which are naturally distinguished based on the environmental sparse reward. A high-level overview of the DuRND framework is illustrated in Figure 1.

## 4.1. Dual Random Network Modules

We introduce two separate RN modules: the positive RN module $\mathcal{R}_P$ and the negative RN module $\mathcal{R}_N$. Each module consists of two networks: a fixed and randomly initialized target network $f_X(s) : S \rightarrow \mathbb{R}^k$, and a differently initialized predictor $\hat{f}_X(s; \theta_X) : S \rightarrow \mathbb{R}^k$, parameterized by $\theta_X$, where $X \in \{P, N\}$. It is worth noting that to prevent estimation bias arising from differences between the two modules, the target networks in $\mathcal{R}_P$ and $\mathcal{R}_N$ are initialized identically, and the predictors in both modules are also initialized identically.

The RN modules, $\mathcal{R}_P$ and $\mathcal{R}_N$, are updated using positive and negative states, respectively, which are identified based on the sparse environmental rewards, as they typically signify task completion or the achievement of a local milestone—both of which strongly align with the agent's goal-directed behaviors. Specifically, the underlying assumption for determining whether a state is positive or negative is as follows: we define the states where original environmental rewards are obtained as *anchor states*. The sequence of states immediately preceding these anchor states is considered positive, as they play a promising role in guiding the agent toward and contributing to achieving the original rewards, while the others are considered negative.

We introduce a hyperparameter $T_{\text{pos}}$, which represents the length of the sequence, i.e., the number of states preceding the anchor states that will be regarded as positive. Furthermore, we propose two approaches to set the hyperparameter $T_{\text{pos}}$: (a) using a **fixed** predefined sequence length, which usually relies on prior knowledge of the environment's reward sparseness or empirical tuning. (b) using an **adaptive** approach to linearly increase the $T_{\text{pos}}$ over training, similar to the concept of the $\epsilon$-greedy strategy. This allows the agent to initially focus on states immediately adjacent to the anchor states and gradually extend the range to include those that occurred earlier or contributed indirectly, thereby expanding the consideration of positive states. The adaptive approach is more flexible and tightly aligns with the smooth transition from exploration to exploitation.

Given a positive or negative state, the corresponding predictor is updated by minimizing the MSE loss:

$$e_X(s_t; \theta_X) = \left\| f_X(s_t) - \hat{f}_X(s_t; \theta_X) \right\|^2, X \in \{P, N\}. \quad (2)$$

By updating the predictors with the observed states, we harness the epistemic uncertainty inherent in deep learning, where error progressively decreases as the volume of training data increases (Burda et al., 2018). Consequently, this error, $e_X$, itself serves as an approximated density estimation for the previously encountered states, with larger errors indicating less frequently visited states, and vice versa.

## 4.2. Novelty and Contribution Rewards

In this section, we introduce how the positive and negative RN modules collaboratively compute two types of rewards: the *novelty reward*, which encourages agents to explore less-visited states; and the *contribution reward*, that guides agents toward states that are more likely to contribute to task completion and maximize environmental rewards.

**Novelty Reward.** Since all historical states are delivered to update either $\mathcal{R}_P$ or $\mathcal{R}_N$, the novelty of a state regarding all previously encountered samples is naturally assessed by combining the prediction errors from both modules, thus the novelty reward is defined as:

$$R^{\text{nov}}(s) = e_P(s) + e_N(s), \quad (3)$$

where $e_P$ and $e_N$ are the prediction errors from $\mathcal{R}_P$ and $\mathcal{R}_N$, respectively, calculated by Equation 2.

By combining the errors from two modules, the novelty reward explicitly evaluates states as three different levels:

1. *High novelty*, both $e_P(s)$ and $e_N(s)$ are high: States that are unseen in both the positive and negative scenarios, indicating a strong need for further exploration.
2. *Medium novelty*, only one of $e_P(s)$ and $e_N(s)$ is high: States observed mainly in one scenario (typically the negative scenario in initial training stages, due to the sparse-reward nature), suggesting that exploration remains encouraged, as states in negative trajectories could potentially become positive in the future.
3. *Low novelty*, both $e_P(s)$ and $e_N(s)$ are low: States encountered in both scenarios, indicating that the agent has already sufficiently explored these states, and additional exploration may not be as beneficial.

Intuitively, this mechanism deliberately adopts an "OR" condition to determine the novelty, meaning a state is considered novel as long as it has not been seen in positive **or** negative scenarios. Compared to the vanilla RND algorithm (Burda et al., 2018), DuRND effectively prolongs the exploration phase, as during the initial training stages, most states are more likely to be classified as negative, the positive RN module maintains a higher error, effectively "holding up" the novelty reward to prevent it from diminishing too quickly, thus encouraging broader and more thorough exploration.

**Contribution Reward.** To evaluate the hidden value of a state, we consider its *positive ratio*, which is defined as the proportion of times a state appears in positive sequences relative to its total historical occurrences (Ma et al., 2025). A higher positive ratio signifies a state's greater likelihood of contributing to achieving environmental rewards, aligning closely with the agent's original objective. Given the prediction errors $e_P(s)$ and $e_N(s)$, which represent the *infrequency* of the corresponding state in their respective scenarios (with their inverses serving as proxies for visiting

frequency), the contribution reward is defined as:

$$R^{\text{con}}(s) = \frac{1/e_P(s)}{1/e_P(s) + 1/e_N(s)} = \frac{e_N(s)}{e_P(s) + e_N(s)}. \quad (4)$$

The contribution reward effectively quantifies the hidden value of a state by estimating the probability of it being positive, providing a statistical measure of its potential to contribute to higher environmental rewards. Notably, due to the non-linear nature of random neural networks, the errors derived from the RN modules are not strictly proportional to the actual visit counts. However, this discrepancy does not undermine our method, as the contribution reward relies on the relative relationship between the positive and negative scales, rather than their absolute values. This design ensures a robust evaluation of a state's contribution, particularly in environments where exact visit frequencies are unavailable or computationally challenging to obtain.

### 4.3. DuRND Enhanced RL Algorithm

In this work, we focus on integrating DuRND into the Proximal Policy Optimization (PPO) algorithm, a well-known advanced on-policy RL algorithm (Schulman et al., 2017). PPO consists of two modules: a policy to select actions given states and a value function to evaluate the policy's behavior. The enhancement is to use the DuRND-defined reward structure in Equation 1 to shape the sparse environmental rewards. Let $\pi_\eta$ be the parameterized policy network and $V_\phi$ be the parameterized Value network. We define the enhanced *advantage* in the DuRND framework as:

$$\hat{A}_t = \sum_{l=0}^{T-t-l} \gamma^l \delta_{t+l}, \quad (5)$$

$$\delta_t = \left(r_t^{\text{env}} + \lambda r_t^{\text{nov}} + \omega r_t^{\text{con}}\right) + \gamma V_{\phi_{\text{old}}}(s_{t+1}) - V_{\phi_{\text{old}}}(s_t).$$

Then the enhanced loss function for policy $\pi_\eta$ is defined as:

$$\hat{L}(\eta) = \mathbb{E}\left[\min\left(r_t(\eta)\hat{A}_t, \text{clip}\left(r_t(\eta),\, 1-\epsilon,\, 1+\epsilon\right)\hat{A}_t\right)\right], \quad (6)$$

where $r_t(\eta) = \frac{\pi_\eta(a_t|s_t)}{\pi_{\eta_{\text{old}}}(a_t|s_t)}$ is the probability ratio, and $\epsilon$ is the clipping parameter. The enhanced loss function for the value function is defined as:

$$\hat{L}(\phi) = \mathbb{E}\left[\left(V_\phi(s_t) - \left(\hat{A}_t + V_{\phi_{\text{old}}}(s_t)\right)\right)^2\right]. \quad (7)$$

By leveraging the novelty and contribution rewards, the augmented DuRND rewards effectively broaden the exploration horizon in early training and reinforce meaningful hidden values in later stages, improving convergence. The trajectory-based optimization nature of PPO fits well with the DuRND's updates. Besides, DuRND, as a standalone

---

**Algorithm 1** DuRND enhanced PPO

**Require:** Environment $\mathcal{E}$, parameterized $\pi_\eta$ and $V_\phi$
**Require:** Random Network modules $\mathcal{R}_P$ and $\mathcal{R}_N$
 1: **for** iteration $i = 1, 2, \ldots$ **do**
 2:    **for** each epoch and $\mathcal{T} = \emptyset$ **do**
 3:       $(s_t, a_t, r_t^{\text{env}}, s_{t+1}) \leftarrow \text{Interact}(\pi_{\eta_{\text{old}}}, \mathcal{E})$
 4:       $e_P(s_t) \sim \mathcal{R}_P,\ e_N(s_t) \sim \mathcal{R}_N$
 5:       $r_t^{\text{nov}} = e_P(s_t) + e_N(s_t)$
 6:       $r_t^{\text{con}} = e_N(s_t)/(e_P(s_t) + e_N(s_t))$
 7:       $\mathcal{T} \leftarrow \mathcal{T} \cup \{(s_t, a_t, r_t^{\text{new}}, r_t^{\text{nov}}, r_t^{\text{con}}, s_{t+1})\}$
 8:    **end for**
 9:    $T_{pos} \leftarrow \text{Schedule}(i)$
10:    **for** $s_\tau \in \mathcal{T}$ **do**
11:       **if** $s_\tau$ is positive: $\mathcal{R}_P \leftarrow \text{Update}(\mathcal{R}_P, s_\tau)$
12:       **else**: $\mathcal{R}_N \leftarrow \text{Update}(\mathcal{R}_N, s_\tau)$
13:    **end for**
14:    $\eta \leftarrow \eta - \alpha_\eta \nabla_\eta \hat{L}(\eta)$          ▷ by Equation 6
15:    $\phi \leftarrow \phi - \alpha_\phi \nabla_\phi \hat{L}(\phi)$          ▷ by Equation 7
16: **end for**

---

reward shaping mechanism, can be easily integrated into various RL algorithms, such as SAC (Haarnoja et al., 2018), TD3 (Fujimoto et al., 2018), and others. We summarize the DuRND-enhanced PPO algorithm in Algorithm 1.

## 5. Experiments

Experiments are designed to evaluate DuRND across various sparse-reward environments. We select twelve challenging tasks from three domains: *Atari*, 2D games from the arcade learning environment (ALE) platform (Bellemare et al., 2013), *VizDoom*, 3D first-person shooting games (Kempka et al., 2016; Tomilin et al., 2022), and *MiniWorld*, simulated 3D maze environments (Chevalier-Boisvert et al., 2023). Specifically, the *MiniWorld* tasks provide rewards only at the end of each episode to indicate task completion, while other tasks offer intermediate rewards for achieving specific milestones, but the overall reward distribution remains highly sparse. All tasks are shown in Figure 2, with the detailed descriptions and the environmental reward structures provided in Appendix A.

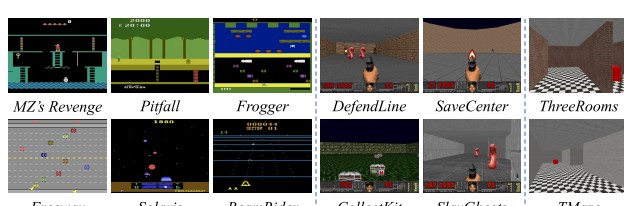

*MZ's Revenge*   *Pitfall*   *Frogger*   *DefendLine*   *SaveCenter*   *ThreeRooms*

*Freeway*   *Solaris*   *BeamRider*   *CollectKit*   *SlayGhosts*   *TMaze*

Figure 2: The sparse-reward tasks in our experiments, including *Atari*, *VizDoom*, and *MiniWorld* domains.

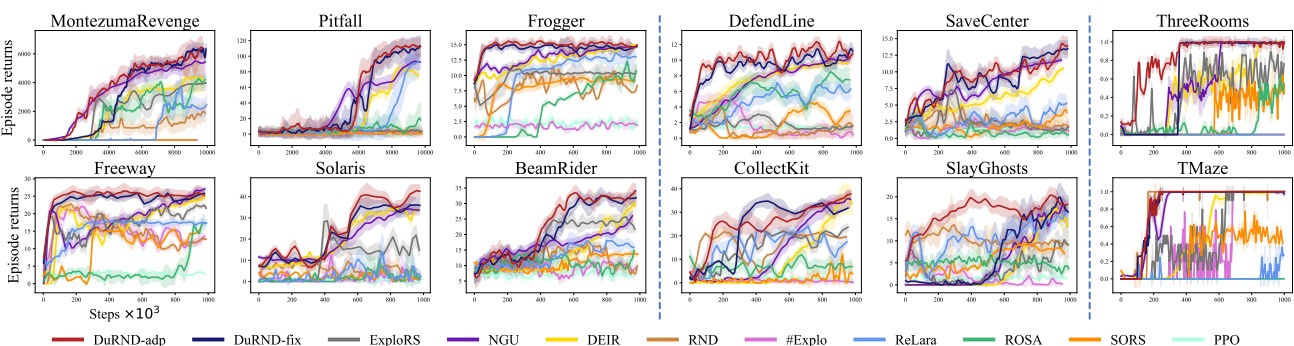

Figure 3: The learning performance of DuRND compared with baselines.

Table 1: Performance comparison of DuRND and baseline models: average episodic returns with standard errors achieved by the trained models, tested over 100 episodes (↑ higher is better).

| Environments | DuRND-adp | DuRND-fix | ExploRS | NGU | DEIR | RND | #Explo | ReLara | ROSA | SORS | PPO |
|---|---|---|---|---|---|---|---|---|---|---|---|
| *MZ'Revenge* | **6654.04±5.26** | 6297.31±1.01 | 3984.74±0.15 | 5459.01±0.08 | 4116.09±0.23 | 1337.34±0.19 | 0.00±0.00 | 2488.06±0.18 | 4443.79±0.45 | 0.00±0.00 | 0.00±0.00 |
| *Pitfall* | 113.15±0.03 | **114.23±0.03** | 4.52±0.00 | 93.02±0.01 | 74.15±0.05 | 4.57±0.00 | 3.51±0.00 | 95.40±0.03 | 14.24±0.04 | 0.69±0.00 | 0.73±0.01 |
| *Frogger* | **14.82±0.00** | 14.67±0.00 | 10.26±0.00 | 14.65±0.00 | 14.80±0.00 | 8.03±0.00 | 2.10±0.00 | 13.02±0.00 | 10.00±0.01 | 9.16±0.00 | 2.27±0.01 |
| *Freeway* | 25.53±0.00 | **25.70±0.00** | 21.93±0.01 | 24.94±0.00 | 24.60±0.00 | 12.80±0.00 | 13.41±0.01 | 17.29±0.00 | 16.81±0.01 | 14.82±0.01 | 3.04±0.00 |
| *Solaris* | **42.63±0.00** | 36.01±0.00 | 18.34±0.06 | 33.74±0.00 | 31.49±0.00 | 4.80±0.02 | 1.47±0.01 | 3.20±0.01 | 2.27±0.02 | 1.38±0.01 | 0.86±0.00 |
| *BeamRider* | **34.13±0.01** | 31.84±0.00 | 21.70±0.02 | 29.21±0.02 | 26.03±0.01 | 9.90±0.02 | 9.33±0.03 | 15.80±0.01 | 14.80±0.01 | 13.64±0.00 | 6.51±0.00 |
| *DefendLine* | **11.03±0.00** | 10.95±0.00 | 1.53±0.00 | 10.22±0.00 | 9.67±0.00 | 1.64±0.00 | 0.39±0.00 | 6.49±0.00 | 7.59±0.01 | 3.40±0.00 | 1.48±0.01 |
| *SaveCenter* | **14.01±0.00** | 13.61±0.00 | 1.73±0.00 | 11.87±0.00 | 10.61±0.00 | 1.87±0.00 | 1.06±0.00 | 5.19±0.00 | 0.50±0.00 | 3.62±0.01 | 1.10±0.00 |
| *CollectKit* | **37.34±0.01** | 31.33±0.01 | 22.50±0.02 | 35.53±0.00 | 30.05±0.01 | 19.68±0.01 | 0.33±0.00 | 17.14±0.02 | 6.25±0.02 | 4.87±0.02 | 1.22±0.00 |
| *SlayGhosts* | **18.29±0.00** | 17.05±0.01 | 10.11±0.02 | 17.29±0.01 | 13.96±0.01 | 9.41±0.00 | 0.26±0.00 | 16.22±0.01 | 3.92±0.01 | 7.76±0.01 | 1.81±0.01 |
| *ThreeRooms* | 0.97±0.00 | **0.99±0.00** | 0.71±0.00 | **0.99±0.00** | 0.97±0.00 | 0.00±0.00 | 0.00±0.00 | 0.00±0.00 | 0.51±0.00 | 0.54±0.00 | 0.00±0.00 |
| *TMaze* | **1.00±0.00** | **1.00±0.00** | **1.00±0.00** | **1.00±0.00** | **1.00±0.00** | **1.00±0.00** | **1.00±0.00** | 0.30±0.00 | 0.00±0.00 | 0.45±0.00 | 0.00±0.00 |

## 5.1. Implementation Details

**Observation Normalization** is a common practice that helps stabilize the learning process. The observations are normalized by subtracting the running mean and dividing by the running standard deviation, following the implementation introduced in (Burda et al., 2018).

**Random Networks Error Normalization.** For different tasks and different initializations of the random network modules, the scale of the MSE errors, $e_S$ and $e_F$, can vary significantly. To formalize the weighting coefficients $\lambda$ and $\omega$ across different tasks, we normalize the MSE errors by dividing them by the *initial error*, which is the average of the MSE errors from the burn-in stage, e.g., 50 episodes before the training process. This is built on the assumption that the errors are gradually decreasing, so the initial error is a good approximation of the scale of the errors. The normalized errors make it possible to set consistent $\lambda$ and $\omega$ for all tasks, avoiding the task-specific tuning.

## 5.2. Comparison with Baselines

We implement two variants of DuRND with different strategies for determining the positive sequence length $T_{\text{pos}}$, as described in Section 4.1. The two variants are denoted as *DuRND-fix* and *DuRND-adp*, where *DuRND-fix* sets $T_{\text{pos}}$ to $1/4$ of the maximum episode length, while *DuRND-adp* lin-

early increases $T_{\text{pos}}$ from 0 to $1/2$ of the maximum episode length over the training. Additional implementation details, including hyperparameters, neural network architectures, and hardware configurations, are provided in Appendix B.

We compare DuRND with eight widely recognized reward shaping baselines, covering the two main categories discussed in this paper. For RS with exploration bonuses, we include *ExploRS* (Devidze et al., 2022), *NGU* (Badia et al., 2020), *DIER* (Wan et al., 2023), *RND* (Burda et al., 2018), and *#Explo* (Tang et al., 2017); For RS with hidden values, we consider *ReLara* (Ma et al., 2024b), *ROSA* (Mguni et al., 2023), and *SORS* (Memarian et al., 2021). Additionally, we compare DuRND with its backbone algorithm, *PPO* (Schulman et al., 2017), to evaluate its performance enhancements. All baselines are implemented using the *RLeXplore* (Yuan et al., 2024), the *CleanRL* (Huang et al., 2022), or the codes provided in the respective papers. To ensure optimal performance, they are configured with either the authors-recommended or well-tuned hyperparameters.

The learning results, averaged over ten runs with different random seeds, are illustrated in Figure 3, while Table 1 presents the average returns achieved by the final model in 100 testing episodes. The DuRND framework demonstrates distinct advantages mainly from three aspects: efficient and directed exploration, rapid and stable convergence, and considerably low training resource demands.

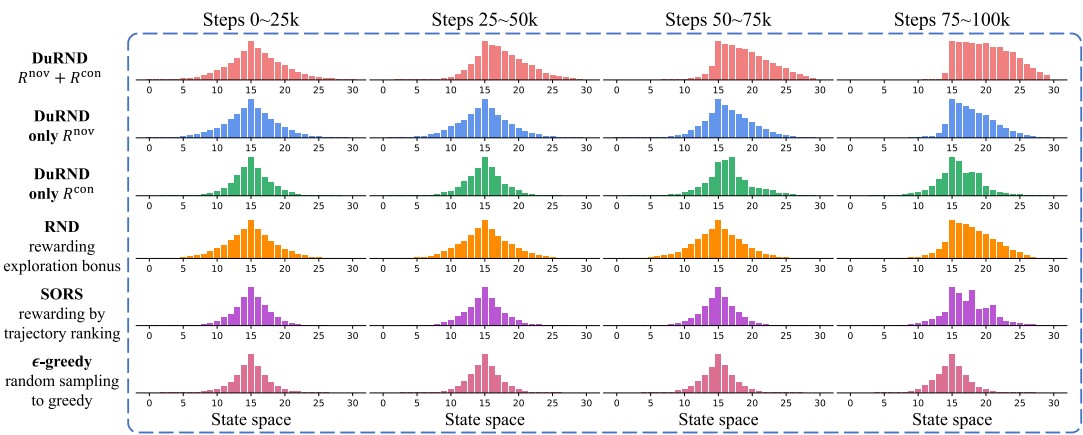

Figure 4: The state visiting distributions of different methods for each 25k steps in the toy task.

**Exploration.** DuRND inherits its exploratory capability from the vanilla RND's strategy (Burda et al., 2018). Rewarding novelty allows the agent to assign higher rewards to less frequently visited states, thus encouraging goal-oriented exploration. For the baselines, ReLara relies on random perturbation on both reward functions and action sampling, which mainly introduces noise to amplify uncertainty; ROSA and SORS depend on the agent's underlying exploration strategies. All three baselines lack explicit guidance on which regions to explore. Consequently, DuRND is observed to collect trajectories with higher episodic returns earlier due to the novelty reward, enhancing sample efficiency. Furthermore, while ReLara, ROSA, and SORS can also converge to optimal policies in many settings, they sometimes remain trapped in local optima. For instance, in the *SaveCenter* tasks, DuRND continuously defeats 12 enemies in one episode, while the baselines only defeat about 6 within the same training periods.

**Exploitation.** The contribution rewards progressively play a more important role, guiding the agent to focus on states that are more likely to yield higher environmental rewards, thereby reinforcing beneficial behaviors. However, for the baselines that only incorporate exploration bonuses, such as ExploRS, RND, and #Explo, agents struggle to derive effective guidance from novelty rewards as training progresses to later stages. Crucially, the agents' overemphasis on novel yet low-value states hinders the recovery from shaping rewards, leading to policies that diverge from the original task objectives. Observations in tasks like *Freeway*, *DefendLine*, and *SlayGhosts* reveal that while these baselines may initially achieve high environmental returns, their performance declines in later stages, deviating from the optimal policies. Conversely, DuRND maintains a steady convergence towards the optimal policy, demonstrating its effectiveness in balancing exploration and exploitation.

**Memory Efficiency.** DuRND is space-efficient as it only

introduces two lightweight RN modules to compute both types of rewards. In comparison, ReLara and ROSA both demand additional agents, which are generally more complex and computationally expensive. ExploRS and #Explo both involve pseudo-counts but are not RND-based, relying instead on density estimations that require substantial extra space for storing all (or at least partial) historical states. To empirically validate DuRND's memory efficiency, we report the maximum memory consumption in Table 2. To provide a more intuitive comparison, we report the relative value normalized to our DuRND. In this case, if the value $> 1$, it indicates that the method is more memory-expensive than DuRND, and vice versa.

Table 2: The maximum memory consumption in three domains, normalized relative to DuRND (↓ lower is better).

| Domains | DuRND | ExploRS | RND | #Explo | ReLara | ROSA | SORS | PPO |
|---|---|---|---|---|---|---|---|---|
| *Atari games* | 1 | 10.94 | 0.91 | 5.72 | 9.61 | 10.14 | 5.29 | 0.83 |
| *VizDoom* | 1 | 11.94 | 0.93 | 5.71 | 9.75 | 10.38 | 5.28 | 0.85 |
| *MiniWorld* | 1 | 11.41 | 0.90 | 5.66 | 9.49 | 10.25 | 5.34 | 0.81 |

### 5.3. Exploration-Exploitation Trade-off

In this section, we further study the exploration-exploitation trade-off in DuRND by demonstrating the differences in state visitation distributions under different reward shaping methods and exploration strategies. For an intuitive illustration, we consider a toy task in a one-dimensional chain of length 31, with states as $s_0, s_1, \cdots, s_{30}$ from left to right. The agent starts at the midpoint, $s_{15}$, at the beginning of each episode. There are 15 states on either side of the starting point, but only the far-right state, $s_{30}$, is the successful terminal state with $R^{\text{env}}(s_{30}) = 1$, while all other states are rewarded as 0. Each episode is limited to a maximum of 20 steps. The agent can take three actions: moving to the left, moving to the right, and staying in the current state.

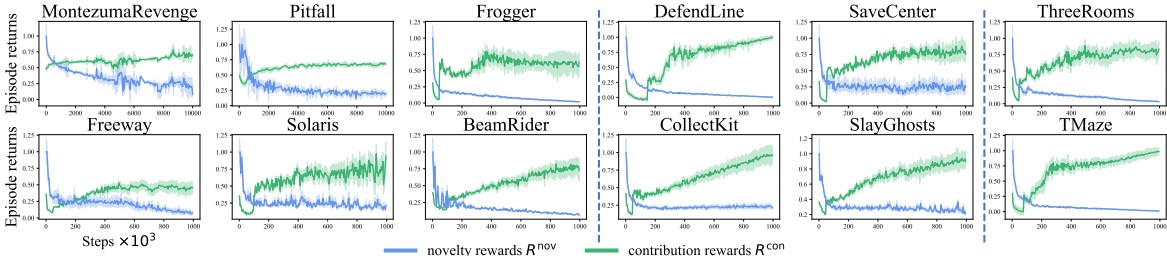

Figure 5: The novelty and contribution rewards learned in the DuRND framework.

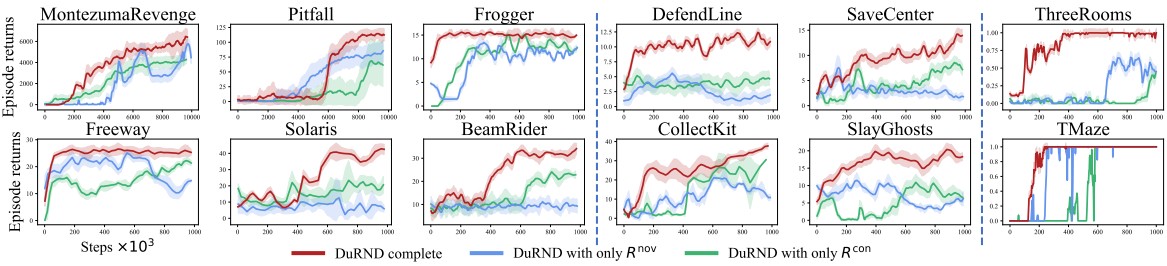

Figure 6: The learning performance of DuRND with a single type of reward.

We compare the complete DuRND-fix ($T_{\text{pos}} = 20$) with two variants: (1) DuRND-fix with only the novelty reward $\lambda R^{\text{nov}}$, and (2) DuRND-fix with only the contribution reward $\omega R^{\text{con}}$; as well as three reward shaping or exploration approaches: (3) *vanilla RND*, that only rewards novelty; (4) *SORS*, that shapes rewards by ranking trajectories with environmental feedback; and (5) $\epsilon$-*greedy*, the popular strategy that selects a random action with probability $\epsilon$ and the greedy action with probability $1 - \epsilon$. For each method, we track the state visitation over a total of 100k steps, presenting the results for every 25k steps in Figure 4.

From the results, we observe that DuRND demonstrates an efficient trade-off between exploration and exploitation. In the early stage (around 0 to 50k steps), DuRND shows a more balanced state visitation across the entire state space, while in the later stage (around 50k to 100k steps), the agent increasingly focuses on the right side of the starting point, as only these states yield positive rewards. In comparison, RND maintains a broader exploratory behavior but is less effective at the exploitation stage, still visiting states on the left side even in the last 25k steps. DuRND with only $\lambda R^{\text{nov}}$ outperforms vanilla RND but falls short of the performance achieved by the complete DuRND, highlighting the effectiveness of the $\omega R^{\text{con}}$ term. SORS and DuRND with only $\omega R^{\text{con}}$ converge more slowly than complete DuRND, and their exploration ranges are more limited. For $\epsilon$-greedy, which lacks a clear exploratory direction, the initial exploration is more concentrated, consequently, it fails to reach the terminal state within the 100k steps.

### 5.4. Novelty and Contribution Rewards

#### 5.4.1. ANALYSIS OF THE LEARNED REWARDS

We discuss how the novelty and contribution rewards evolve during training. Figure 5 shows the normalized rewards received by the agent throughout learning. Over time, the novelty reward decreases while the contribution reward increases, both nonlinearly. The decline in the novelty reward indicates the diminishing differentiation among states after extensive exploration, i.e., states become uniformly non-novel, thus, the information provided by novelty rewards loses significance in later training, highlighting again the limitation of relying only on novelty may hinder convergence. The contribution reward increases and eventually stabilizes at a high level, dominating the shaping rewards. This is attributed to the continuous reinforcement of positive states, which leads to a gradual decrease in $e_P$, causing the contribution reward $R^{\text{con}} = e_N/(e_N + e_P)$ to approach 1 and stabilize at a consistent level. In summary, the transition from exploration-driven to task-oriented rewards is a critical factor underpinning DuRND's superior performance.

#### 5.4.2. EFFECTS OF TWO REWARDS

To further understand the effects of two types of rewards, we compare the complete DuRND-adp with two variants: (1) DuRND-adp with only the novelty reward (*only $R^{\text{nov}}$*), and (2) DuRND-adp with only the contribution reward (*only $R^{\text{con}}$*). The learning curves are shown in Figure 6, with the quantitative results in Appendix C.3.

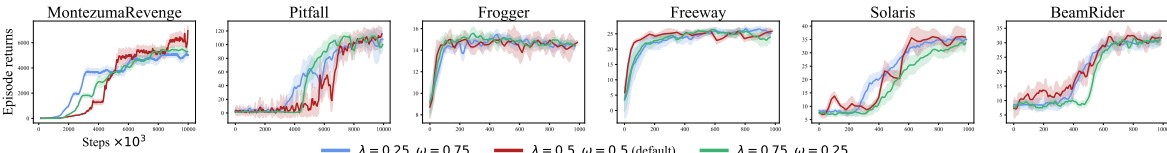

Figure 7: Ablation study: learning performance of DuRND with different positive sequence length $T_{\text{pos}}$.

Figure 8: Ablation study: the learning performance of DuRND with different reward coefficients.

The results show that both rewards are essential for DuRND. When relying only on the novelty reward, agents struggle to recover the environmental rewards, leading to unstable convergence and deviations from the task's original objectives. However, it is worth noting that this variant outperforms the vanilla RND, due to the two RN modules explicitly defining three levels of novelty as described in Section 4.2, which effectively enforces the agent to explore potentially valuable states, thereby expanding the scope of exploration. In contrast, using only the contribution reward hinders efficient exploration, delaying favorable outcomes, and potentially trapping the agent in local optima.

### 5.5. Ablation Study

We conduct ablation studies to analyze the effects of two key components in DuRND: the positive sequence length $T_{\text{pos}}$ and the reward coefficients $\lambda$ and $\omega$, in the Atari games.

**The positive sequence length** $T_{\text{pos}}$ (Figure 7). We observe that the adaptive strategy to address $T_{\text{pos}}$ generally leads to better performance, as it allows a smooth transition from exploration to exploitation. However, DuRND is robust to the choice of $T_{\text{pos}}$, even with different fixed values, the agent can still achieve satisfactory performance.

**The reward coefficients** $\lambda$ **and** $\omega$ (Figure 8). It is observed that DuRND is not sensitive to the choice of $\lambda$ and $\omega$, if they are set within a reasonable range. But it still controls which reward plays a more dominant role in the shaping rewards, thus affecting the exploration-exploitation trade-off. A higher $\lambda$ leads to more exploration, while a higher $\omega$ leads to earlier exploitation. Therefore, a balanced setting of $\lambda$ and $\omega$ is recommended for optimal performance.

## 6. Discussion and Conclusion

**Conclusion.** This paper introduces DuRND, a framework that separately estimates state visitation frequencies in posi-

tive and negative scenarios. The dual RN modules compute two types of rewards, achieving both directed exploration and stable convergence. Experiments demonstrate that, unlike novelty-based RS methods, DuRND avoids the pitfalls of continuously novelty-driven exploration, instead shifting to more meaningful rewards for desired behaviors; while compared to hidden value based RS methods, it broadens the exploration and collects more diverse samples. In summary, DuRND combines the strengths of both approaches, achieving an efficient balance between exploration and exploitation. Lastly, DuRND operates with low computational overhead in high-dimensional environments, making it a scalable solution for a wide range of complex RL tasks.

**Limitations.** As an RS method designed for sparse-reward tasks, DuRND relies on environmental rewards to distinguish positive and negative states. In dense-reward tasks, where such distinctions are less clear, its effectiveness may be limited. Additionally, the hyperparameter $T_{\text{pos}}$, was implemented with fixed and adaptive approaches. While both methods are intuitive and demonstrated considerable robustness in our experiments – achieving consistent results across different environments using the same configuration – a more sophisticated design that autonomously adjusts $T_{\text{pos}}$ based on reward sparsity could further enhance adaptability.

## Acknowledgement

This research is supported by Academic Research Grants MOE-T2EP20121-0015 and MOE-T1 251RES2408 from the Ministry of Education in Singapore.

## Impact Statement

This paper presents work whose goal is to advance the field of Machine Learning. There are many potential societal consequences of our work, none of which we feel must be specifically highlighted here.

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

# A. Environments Configuration

All tasks in our experiments provide sparse rewards. The objective descriptions and the criteria for assigning sparse environmental rewards are detailed in Table 3. Apart from tasks *ThreeRooms* and *TMaze*, which offer episodic rewards, other tasks provide intermediate rewards upon the completion of some specific milestones. All other states yield zero rewards.

Table 3: Objective descriptions and environmental rewards assignments for the tasks in our experiments.

| Environments | Sparse Rewards Assignment |
|---|---|
| *MZ's Revenge* | Control the explorer to navigate through a series of rooms, avoiding traps and enemies while collecting items.
1. Rewards are given for collecting items such as keys, treasures, and other objects.
2. Rewards vary depending on the item collected, e.g., $+100$ for picking up a key.
3. Higher rewards are given for unlocking doors or accessing new areas, often in the range of $+300$ to $+1000$.
4. No reward is given for simply traversing rooms or surviving; rewards are sparse and tied to specific achievements.
5. Episode ends when all lives are lost or maximum steps 4000 are reached. |
| *Pitfall* | Control the explorer through a jungle environment to collect treasures while avoiding obstacles.
1. $+2$ reward for collecting a gold bar.
2. $+4$ reward for collecting a diamond ring.
3. $+5$ reward for collecting a silver bar.
4. No rewards are given for surviving obstacles like pits, crocodiles, or rolling logs.
5. Negative rewards (e.g., $-1$) are incurred for falling into pits or touching harmful objects.
6. Episode ends when time runs out (maximum steps 2000) or all treasures are collected. |
| *Frogger* | Guide the frog home across a highway and river while avoiding cars and predators.
1. $+2$ rewards for reaching home.
2. $+1$ reward for eating a fly.
3. Episode ends when all 5 frogs are lost or maximum steps 2000 are reached. |
| *Freeway* | Guide the chicken across multiple lanes of heavy traffic.
1. $+1$ reward for the chicken goes across the screen.
2. Episode ends if all 3 chickens are hit by cars or maximum steps 2000 are reached. |
| *Solaris* | Control a spaceship to blast enemies and explore new galaxies.
1. $+1$ reward for destroying a target.
2. $+1$ reward for entering a new galaxy.
3. Episode ends when all ships are destroyed or maximum steps 2000 are reached. |
| *BeamRider* | Control a spaceship to destroy enemies while avoiding obstacles.
1. $+1$ reward for each enemy ship destroyed.
2. Episode ends if all ships are lost or maximum steps 2000 are reached. |
| *DefendLine* | Defend the line by neutralizing incoming enemies.
1. $+1$ reward for each enemy killed.
2. Episode ends if the player is defeated or the maximum steps 1000 are reached. |
| *SaveCenter* | Protect the center by eliminating enemies.
1. $+1$ reward for each enemy killed.
2. Episode ends if the player is defeated or the maximum steps 1000 are reached. |
| *CollectKit* | Collect health kits in a room full of poison.
1. $+1$ reward for collecting one kit.
2. Episode ends if the player is killed by the poison or the maximum steps 1000 are reached. |
| *SlayGhosts* | Eliminate ghosts or monsters in a designated environment.
1. $+1$ reward for each ghost killed.
2. Episode ends if the player is killed or the maximum steps 1000 are reached. |
| *ThreeRooms* | Navigate through three connected rooms to reach a red cube.
1. $+1$ reward for reaching the red cube.
2. $-0.1$ penalty for each time step taken.
3. Episode ends when the cube is reached or the maximum steps 500 are reached. |
| *TMaze* | Navigate a T-shaped maze to reach the red cube.
1. $+1$ point for reaching the red cube.
2. $-0.1$ penalty for each time step taken.
3. Episode ends when the cube is reached or the maximum steps 500 are reached. |

# B. Experiments Implementation Details

### B.1. Hyperparameters

DuRND is relatively robust to hyperparameters, we report the hyperparameters used in our experiments in Table 4.

Table 4: The hyperparameters of DuRND in our experiments.

| Hyperparameters | Values |
|---|---|
| discount factor $\gamma$ | 0.99 |
| generalized advantage estimate (GAE) coefficient | 0.95 |
| rollout length | 2048 |
| burn-in and error normalization episodes | 50 |
| number of mini-batches | 32 |
| number of update epochs | 10 |
| learning rate | $3 \times 10^{-4}$ |
| maximum gradient normalization | 0.5 |
| random networks learning rate | $10^{-6}$ |
| PPO clip coefficient | 0.2 |
| PPO value loss coefficient | 0.5 |
| novelty reward weight $\lambda$ | 0.5 |
| contribution reward weight $\omega$ | 0.5 |
| DuRND-fix length of the positive sequence $T_{\text{pos}}$ | 1/4 of the episode length |
| DuRND-adp minimum (initial) positive sequence $T_{\text{pos}}$ | 1 |
| DuRND-adp maximum (end) positive sequence $T_{\text{pos}}$ | 1/2 of the episode length |

### B.2. Neural Network Architectures

The neural network architecture of the PPO agent used in our experiments is shown in Figure 9. The architecture of the random network is shown in Figure 10.

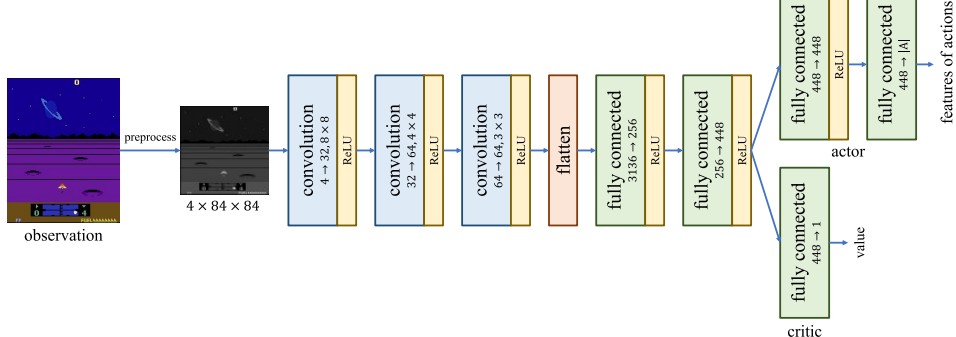

Figure 9: The neural network architecture of the PPO agent in our experiments.

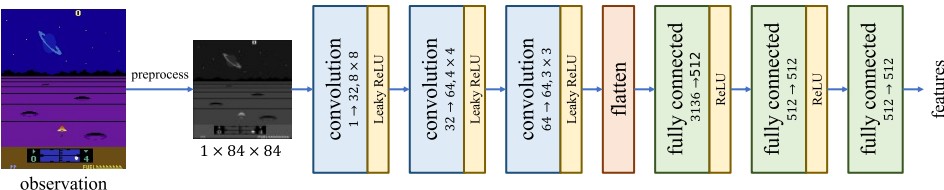

Figure 10: The neural network architecture of the random network in our experiments.

### B.3. Hardware Configurations

The experiments are conducted on machines mainly with two kinds of configurations:

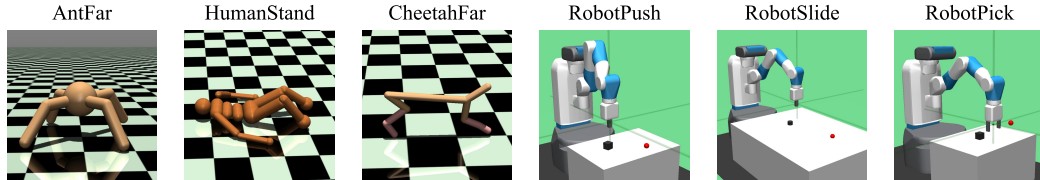

Figure 11: The six continuous-control tasks used in our experiments, covering both *MuJoCo* and *robotics* domains.

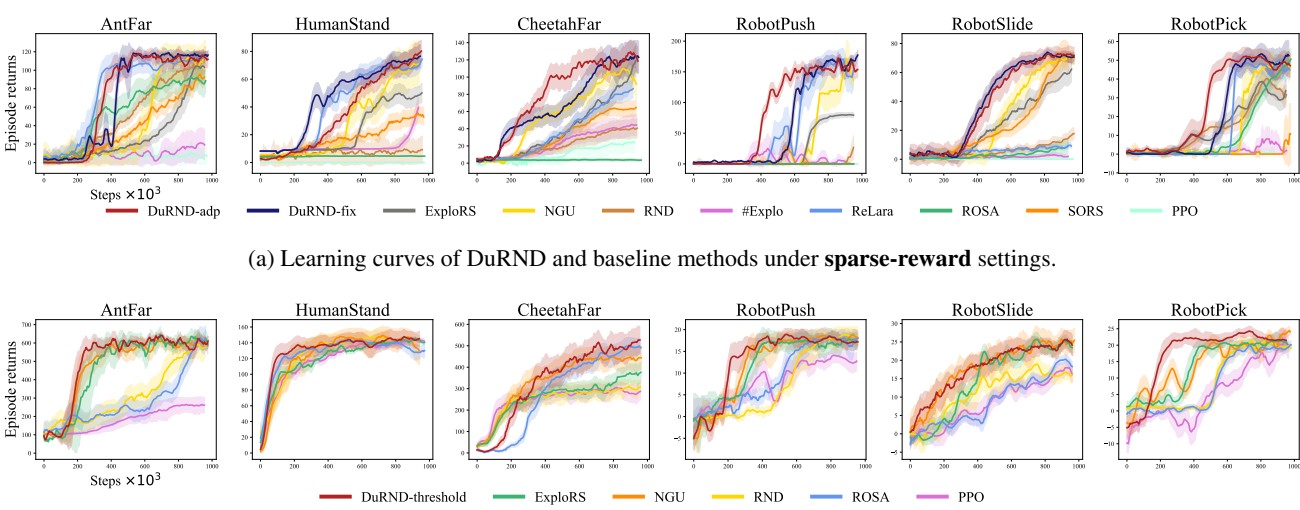

(a) Learning curves of DuRND and baseline methods under **sparse-reward** settings.

(b) Learning curves of DuRND and baseline methods under **dense-reward** settings.

Figure 12: Evaluation of DuRND on continuous-control tasks in both sparse and dense-reward settings.

1. The GPU is NVIDIA Tesla A100 with 40GB memory. The CPU is Intel Xeon Gold 6326 with 16 cores and 32 threads.
2. The GPU is NVIDIA Tesla H100 with 40GB memory. The CPU is AMD Epyc 9334 with 32 cores and 64 threads.

The experiments are implemented by *PyTorch* in version 2.0.1 and *CUDA* in version 11.7.

## C. Additional Experimental Results

### C.1. DuRND for Dense-Reward Scenarios

DuRND is able to be extended to dense-reward scenarios, by slightly modifying the positive sequence length $T_{\text{pos}}$ mechanism. In dense-reward scenarios, we set the *anchor states* as the states with a reward greater than a certain threshold, which is set to $0.5$ in our experiments. Following this modification, we conduct experiments on six continuous-control tasks in the *MuJoCo* and *robotics* domains, shown in Figure 11, evaluating under both sparse and dense-reward settings. The results are shown in Figure 12, Tables 5 and 6.

Table 5: Performance evaluation in the continuous-control tasks with **sparse-reward** setting: average episodic returns with standard errors achieved by the trained models, tested over 100 episodes. (↑ higher is better).

| Tasks | DuRND-adp | DuRND-fix | ExploRS | NGU | RND | #Explo | ReLara | ROSA | SORS | PPO |
|-------|-----------|-----------|---------|-----|-----|--------|--------|------|------|-----|
| *AntFar* | 111.07±0.03 | **116.17±0.05** | 107.34±0.11 | 111.42±0.01 | 110.21±0.10 | 17.92±0.06 | 112.86±0.02 | 91.40±0.09 | 91.39±0.05 | 6.82±0.04 |
| *HumanStand* | **82.60±0.04** | 77.56±0.02 | 50.37±0.03 | 69.54±0.05 | 8.50±0.01 | 42.87±0.05 | 76.09±0.01 | 4.58±0.00 | 33.48±0.03 | 0.03±0.00 |
| *CheetahFar* | **124.18±0.05** | 123.11±0.06 | 119.01±0.11 | 81.59±0.02 | 39.60±0.05 | 42.35±0.03 | 90.53±0.02 | 3.59±0.01 | 68.68±0.06 | 22.69±0.05 |
| *RobotPush* | 173.35±0.04 | **177.20±0.04** | 78.46±0.01 | 119.32±0.05 | 26.27±0.18 | 0.03±0.00 | 170.16±0.04 | 0.00±0.00 | 0.00±0.00 | 0.00±0.00 |
| *RobotSlide* | **71.88±0.02** | 70.65±0.01 | 67.07±0.04 | 69.86±0.02 | 17.78±0.01 | 2.01±0.01 | 8.35±0.02 | 11.51±0.05 | 65.79±0.05 | 0.30±0.00 |
| *RobotPick* | 49.32±0.03 | **51.50±0.04** | 38.44±0.05 | 46.27±0.03 | 29.54±0.10 | 3.42±0.01 | 41.26±0.01 | 47.67±0.02 | 14.05±0.10 | 0.90±0.01 |

Table 6: Performance evaluation in the continuous-control tasks with **dense-reward** setting: average episodic returns with standard errors achieved by the trained models, tested over 100 episodes. (↑ higher is better).

| Tasks | **DuRND-threshold** | ExploRS | NGU | RND | ROSA | PPO |
|---|---|---|---|---|---|---|
| *AntFar* | **618.52±0.18** | 561.19±0.42 | 589.16±0.06 | 582.31±0.12 | 613.19±0.64 | 260.95±0.07 |
| *HumanStand* | **144.56±0.03** | 141.74±0.01 | 130.53±0.06 | 141.53±0.05 | 128.93±0.03 | 142.49±0.02 |
| *CheetahFar* | **531.24±0.17** | 367.50±0.37 | 474.59±0.20 | 314.77±0.18 | 486.08±0.25 | 291.26±0.05 |
| *RobotPush* | 17.42±0.00 | 17.37±0.02 | **19.07±0.01** | 16.99±0.01 | 18.11±0.01 | 13.40±0.01 |
| *RobotSlide* | **23.48±0.01** | 22.73±0.01 | 22.45±0.01 | 15.75±0.02 | 17.66±0.00 | 15.03±0.01 |
| *RobotPick* | 21.27±0.01 | 20.14±0.01 | **23.53±0.01** | 18.98±0.00 | 20.38±0.01 | 17.62±0.04 |

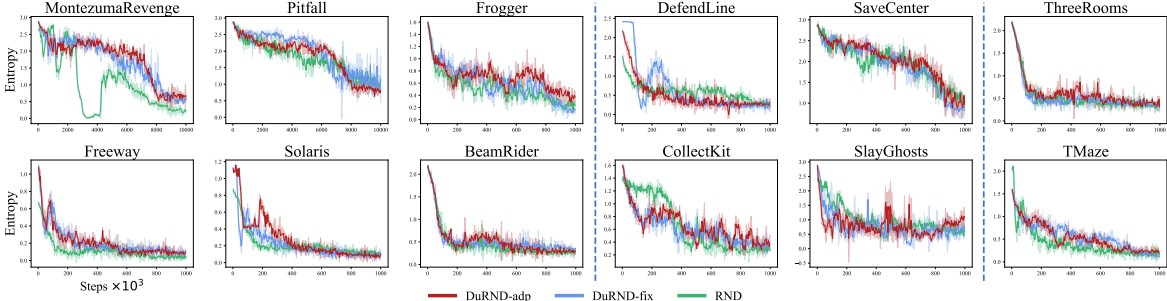

Figure 13: Comparison of policy entropy in DuRND-adp, DuRND-fix, and RND throughout training.

## C.2. Entropy Comparison

To further analyze the behavior of the policy, we compare the policy entropy of DuRND-adp, DuRND-fix, and RND throughout training. The results are shown in Figure 13, which reveals: (a) Entropy in both methods drops in similar period (around 30% of training) with similar rates; (b) In early stage, DuRND maintains higher entropy, especially in complex tasks, indicating higher action diversity and broader exploration.

## C.3. Quantitative results for the Effects of Two Rewards

To support the main results of the study in our paper, we provide quantified results of DuRND with a single type of reward in Table 7. The results show that both types of rewards are essential for the DuRND framework to achieve the best performance.

Table 7: Performance comparison of DuRND-adp with a single type of reward: average episodic returns with standard errors achieved by the trained models, tested over 100 episodes (↑ higher is better).

| Environments | DuRND | DuRND with only $R^{con}$ | DuRND with only $R^{nov}$ |
|---|---|---|---|
| *Montezuma's Revenge* | 6297.31±1.01 | 4300.24±1.65 | 4325.56±1.29 |
| *Pitfall* | 113.15±0.03 | 60.71±0.03 | 87.62±0.04 |
| *Frogger* | 14.82±0.00 | 11.95±0.00 | 12.11±0.00 |
| *Freeway* | 25.53±0.00 | 21.62±0.01 | 14.76±0.00 |
| *Solaris* | 42.63±0.00 | 19.09±0.02 | 6.40±0.01 |
| *BeamRider* | 34.13±0.01 | 22.85±0.02 | 9.28±0.00 |
| *DefendLine* | 11.03±0.00 | 4.50±0.00 | 1.82±0.00 |
| *SaveCenter* | 14.01±0.00 | 6.91±0.00 | 1.69±0.00 |
| *CollectKit* | 37.34±0.01 | 28.78±0.03 | 10.95±0.00 |
| *SlayGhosts* | 18.29±0.00 | 6.70±0.00 | 5.90±0.01 |
| *ThreeRooms* | 0.97±0.00 | 0.39±0.00 | 0.46±0.00 |
| *TMaze* | 1.00±0.00 | 1.00±0.00 | 1.00±0.00 |

