# OpenReview forum: "Catching Two Birds with One Stone: Reward Shaping with Dual Random Networks for Balancing Exploration and Exploitation"
_ICML.cc/2025/Conference — ICML 2025 poster_

### Official Review · Reviewer_GFgX · 2025-03-08

**Overall Recommendation:** 3

**Summary:**

This work develops a new reward shaping approach “DuRND” specialized for sparse reward environments, which uses two random networks (RNs): one RN guides agents to goal states while the other RN prevents the agent from getting stuck in distracting or harmful states. The method is tested against other reward shaping methods in a variety of environments, with the learned reward signals and high-level exploration analyzed by additional experiments.

**Claims And Evidence:**

From “main contributions”:

**Claim:**  DuRND achieves exploration-efficient and convergence-stable learning in challenging sparse-reward tasks.

**Evidence:** Figure 3,5,6 supports this claim.

**Claim:** DuRND is lightweight and highly scalable (in high-dimensional environments).

**Evidence:** Table 2 supports this claim.

**Claim:** The effectiveness and efficiency of DuRND are validated across a variety of sparse-reward tasks with high-dimensional states, demonstrating its superior performance compared to several benchmarks.

**Evidence:** Figure 3 supports this claim.

Other claims stated in the paper:

**Claim:** “However, both branches require an environmental transition model, which makes them challenging in adapting to large-scale scenarios with complex dynamics”

**(Lacking Evidence):** This claim does not seem accurate, and needs backed up. In potential-based reward shaping, the next-state is indeed required to calculate $\Phi(s’)$, but this is always given in the (SARS') mini-batch and does not require a dynamics model.
Can you please explain further, or validate or adjust the phrasing of the claim?

**Essential References Not Discussed:**

Baselines to consider (low memory, work on dense reward also). If unable to provide further experiments, could you please discuss the relationship of your method to these?
- https://arxiv.org/abs/2107.08888
- https://arxiv.org/pdf/2412.01114
- https://arxiv.org/abs/2501.00989

Papers for a broader audience:
- https://arxiv.org/pdf/2408.10215
- https://arxiv.org/pdf/2409.05358

**Experimental Designs Or Analyses:**

The experiments are sound; testing on well-established sparse reward environments. However, the chain environment can benefit from a figure and/or reference based on its previous use. I am also somewhat concerned about the performance of the RND baseline. Upon referring to the paper https://arxiv.org/pdf/1810.12894, I see much higher returns for Montezuma and Pitfall. Could you please explain the discrepancy? I also see that e.g. ReLara was tested only on Mujoco/robotics environments, not any of your chosen benchmarks. Could you explain if there is anything stopping DuRND from performing well on such environments?

**Methods And Evaluation Criteria:**

Do proposed methods and/or evaluation criteria (e.g., benchmark datasets) make sense for the problem or application at hand?

Yes. The environments are all suitable choices for sparse reward settings.

**Other Comments Or Suggestions:**

- Can you discuss or define "hidden values" since it is mentioned early in the paper?
- Is the "N" network (Eq. 2) still trained before a reward is observed? Or do we have to wait to observe a reward before training the networks? Can this lead to inefficiencies?
- Can condense discussion after Eq 3 and move table 1 to appendix in exchange for enlarging Fig  3 (split into two bigger figures)
- typo: "libarary" L312
- "Explotation" subsection p6, can you connect this discussion to Equation 4?
- Fig 3 caption: what are shaded regions?
- Fig 4 caption: which env?
- typo: section 6, "DuRND." to "DuRND,"
- typo: should be $\gamma \in [0,1)$

**Other Strengths And Weaknesses:**

Strengths:

Although I am not entirely familiar with the relevant literature, this appears to be a novel and creative use of RND that has a significant impact on performance. The illustrative example in Fig 4 clearly provides an intuition for the algorithm's effect. Importantly, the method does not require considerable memory nor additional data.


Weaknesses:

There is room for further improvement and understanding on the theoretical and algorithmic sides. Algorithmically, further ablation studies (e.g. on $\lambda, \omega, T_{pos}$) and tests on dense reward environments would have helped gain a broader perspective of performance.

"Other":

(Both strength and weakness) Though it is not a theoretical paper, some further grounding in the choices for reward functions could help: Section 5.3.1 is quite helpful here, but can we further understand the asymptotic reward values? What about the relationship to the extrinsic env reward scale? How about the effect of $T_pos$ and its relationship to important MDP parameters like mixing time? On the flip side, I think these questions can open the door for interesting future work.

**Questions For Authors:**

- What if in sparse reward setting, some rewards are negative? How does this impact the discussion after L170?
- Why is a bigger $T_{pos}$ better for exploitation?
- You mention combining DuRND with SAC; do you have any intuition on how does the MaxEnt "exploration/entropy bonus" affect/interact your method?
- L312, how did you tune such hyperparameters?
- What's the difference between (1) and (3) in section 5.2?
- Since the shaping method is not PBRS, is this method guaranteed to find the optimal policy? Can you please comment on this caveat in the main text? Perhaps in the limit, if one can show the auxiliary rewards tend to zero, such a guarantee can be made.

**Relation To Broader Scientific Literature:**

This paper offers a new method for navigating exploration-exploitation in sparse settings. Typically, RS methods in sparse environments focus solely on exploration, which can explain the performance improvement. The use of two RNDs and the corresponding construction of two reward signals seems novel and interesting. The use of this method however, may not be plug-and-play with any RL approach, specifically in dense reward environments, as this was not tested. The idea connects to a similar line of thought in recent RS research, including SORS, ROSA, ReLara and RND.

**Theoretical Claims:**

N/A

It does not seem there are any theoretical claims needing substantiation. Providing additional theoretical insights to the method could strengthen the work, though.

---

> ### Author Rebuttal · Authors · 2025-04-01
>
> Dear reviewer,
>
> Thanks for the valuable comments. We respond as follows.
>
> ---
>
> Regarding the claim on PBRS, thanks for pointing this out, our statement may cause confusion, many PBRS methods don't require a dynamic model, instead compute potentials directly by collected data. What we intended to highlight is that many information gain-based methods rely on explicit models (e.g., train forward models to predict state and compare with observed state). We'll clarify this.
>
> ---
> Regarding comments on experiments, i.e., baselines, continuous-control, dense-reward, and ablation studies:
> 1. **RND baseline discrepancy**: The gap is mainly due to training budget: (a) the best results reported in RND paper used 1B steps and 1024 parallel envs (from Appendix A.4 and official codes). Due to limited resources, we ran ~1/4 of their full training steps with 32 parallel envs. Our reproduced results align with RND paper's Figure 6 (left), achieving ~2k return in MZ's Revenge. However, with the same reduced training steps, DuRND achieves ~6k, also able to show the improvement.
> 2. **ReLara's continuous-control benchmarks, and dense-reward tasks**: The intention we focused on Atari, VizDoom and 3DMaze is to highlight the advantage of DuRND for image-based high-dimensional states. We agree that MuJoCo and robotics in ReLara are important benchmarks, and also for reviewer's comment on dense-reward tasks, we added [experiments on 6 new tasks, covering both sparse and dense reward settings (Fig A.2, Table A.2 A.3) [link]](https://anonymous.4open.science/api/repo/ano/file/2.pdf). DuRND can be naturally extended to dense rewards by a thresholding strategy: states will $R^{env}$ exceeds a threshold (0.5 in implementation) is recorded by positive RN; otherwise negative RN. Results show DuRND is robust in continuous-control and dense-reward tasks, validating generality.
> 3. **More ablation studies**: We extended ablations on: (a) [the weights $\lambda$, $\omega$ (Fig A.3) [link]](https://anonymous.4open.science/api/repo/ano/file/3.pdf) and (b) [the $T_{\text{pos}}$ (Fig A.4) [link]](https://anonymous.4open.science/api/repo/ano/file/4.pdf). Results show DuRND remains stable under varied settings.
>
> All new experiments will be added in paper.
>
> ---
> We'll move appendix B.1 to main text.
>
> ---
> Regarding suggested related works:
> 1. Yuan et al. (MMRS): count-based exploration that treats number of visits as a limited resource and uses Jain’s fairness index to allocate visits to states, prioritizing under-visited states.
> 2. Koprulu et al.: PBRS method constructing potential from both task-agnostic prior experience and task-specific expert demonstrations, addressing distribution mismatch.
> 3. Adamczyk et al. (BSRS): PBRS method using agent’s current estimate of value function as potential, provided convergence guarantees.
> 4. Ibrahim et al.: survey on reward engineering and shaping.
> 5. Lidayan et al. (BAMDP): Unifies intrinsic motivation and PBRS, ensuring convergence to the original optimal policy under BAMAP framework.
>
> They are relevant to RS and will be properly discussed and cited in paper.
>
> ---
> Other Comments response
> - OC1. "Hidden values" mean state's latent importance that not reflected in env rewards, e.g., in maze tasks, keys or doors carry high value compared to other states
> - OC2. RN modules update after each rollout. Once episode ends or $R^{env}$ is observed, states in rollout buffer are devided to $R_P$ or $R_N$. The delay is bounded by rollout length and doesn't affect efficiency
> - OC7. Shaded areas: standard error over 10 seeds
> - OC8. Figure 4: The toy task in Section 5.2
>
> We'll fix remaining comments (typos, captions, layout).
>
> ---
> Question response
> - Q1: negative rewards usually indicate undesirable states, which DuRND naturally assigns to negative RN
> - Q2: Larger $T_{pos}$ classifies more states as positive, potentially increasing $R^{con}$ coverage. But overly large $T_{pos}$ may mislabel irrelevant states. The effect is non-monotonic and requires balance
> - Q3: Current DuRND has actually used the entropy in PPO backbone (by CleanRL), which encourages action diversity. While shaped rewards guide directional exploration, they are not conflicting. Further analysis is a promising direction
> - Q4: We mostly follow author-recommended hyperparameters from original papers or official codes, with a small set tuning. Results are averaged over 10 seeds
> - Q5: DuRND with only $R^{nov}$ retains the dual-RN structure, standard RND used a single RN. It preserves three-level novelty and enables broader exploration
> - Q6: DuRND is not PBRS, thus doesn't theoretically guarantee the original optimal policy, but empirically converges to high-performing policies. In Fig 5, $R^{nov}$ often diminishes while $R^{con}$ remains active, but since $R^{con}$ aligns with task rewards, DuRND has a high likelihood of converging to original optimal policy. We'll add this discussion.
>
> Thanks again for the comments and hope our response has addressed your concerns.

---

### Official Review · Reviewer_Yo5a · 2025-03-10

**Overall Recommendation:** 4

**Summary:**

This paper proposes DuRND, a simple variation on top of RND that uses two random networks in order to compute two reward bonus terms for sparse reward tasks: 1) a modified novelty bonus and 2) an exploitative reward shaping bonus. The two random networks are trained on different data, with the positive network only trained on states that lead up to the sparse reward, while the negative network is trained on all states. Both reward bonus terms summed up is able to better adapt to sparse reward tasks as well as automatically do the exploration-exploitation tradeoff better. Experimental results show that DuRND is consistently better than RND and other related baselines.

## Update after Rebuttal

I maintain my positive score on this paper.

**Claims And Evidence:**

- The claims made are mostly supported by empirical results and ablations.

- A minor but important clarification is that the novelty bonus computed from the two RNs depends on the data being more or less on-policy. If we imagine using an off-policy algorithm like DQN, where we use a replay buffer, but we also use e-greedy noise where we actually run a family of policies each with different epsilons (noise levels), and don’t decay the epsilons (this was the case for Agent57), then in theory the novelty bonus won’t go away completely. This is because there will always be some high epsilon-greedy data in the replay buffer that will be sampled, which may never see the “goal”, and thus never be labeled as positive states. Then the positive RN will never reduce uncertainty on those states, and thus you will get an irreducible novelty signal. For the original RND, as well as other pseudo-count or ensemble based methods, they don’t suffer this issue because they see all the data (like the negative RN). Since the paper uses an on-policy algorithm, this doesn’t come up, but it is important to clarify.

**Essential References Not Discussed:**

Seems like the most essential ones are covered.

**Experimental Designs Or Analyses:**

The overall description of the experiments make sense.

**Methods And Evaluation Criteria:**

Yes domains and ablations make sense.

**Other Comments Or Suggestions:**

In Figure 6, the ablation with each separate novelty bonus, I would like to also see (either in this plot, or a new plot in the appendix) two more baselines like RND and Relara as it would just be easier to compare all on one plot rather than scrolling back and forth.

**Other Strengths And Weaknesses:**

As mentioned by the paper, the approach is designed for sparse reward tasks and not for dense reward tasks, which limits the applicability of the approach to many continuous control and robotics tasks.

**Questions For Authors:**

In Pitfall, RND flatlines, but DuRND with just novelty is actually able to take off quite well. In Montezuma’s revenge, DuRND with just novelty is also doing better than RND. Outside of these two domains, it seems like DuRND with just novelty and RND are similar. I wonder if the authors have any detailed insight into why DuRND with just novelty is much better than RND for these two domains? The current explanation of the 3 levels of novelty is just a hypothesis, and it would be great if see where the difference actually is. In Figure 4, from the state visitation plots, RND and DuRND with just novelty basically look identical.

**Relation To Broader Scientific Literature:**

The algorithm builds upon RND and the reward shaping literature in a simple but nice way.

**Theoretical Claims:**

n/a

---

> ### Author Rebuttal · Authors · 2025-04-01
>
> Dear reviewer,
>
> Thank you for your valuable feedback. We respond to your comments as follows:
>
> ---
>
> Regarding the reviewer’s insightful comment on the interaction between novelty estimation and data distribution under off-policy settings, as noted, since DuRND is built on an on-policy algorithm (PPO), the dual RN modules receive data from a consistent policy distribution, and the labeling of positive/negative states remains stable and accurate throughout training. This prevents the kind of irreducible novelty bias described in off-policy settings. We appreciate this important observation and will clarify the current on-policy assumption and its implications for the RN-based novelty estimation in the revised paper.
>
> ---
>
> Regarding the concern about applicability to continuous-control and dense-reward tasks:
>
> 1. First, DuRND is fully compatible with continuous-control environments, as it is a relatively independent module that can be seamlessly integrated into continuous-action backbone algorithms.
> 2. Second, although DuRND is designed for sparse-reward challenge, it can be naturally extended to dense-reward settings. We adopt a simple yet effective threshold-based strategy: if a state's environmental reward exceeds a predefined threshold (0.5 in implementation), it is recorded in the positive RN; otherwise, negative RN. We evaluated this extension on [six continuous-control tasks across MuJoCo and Robotics domains, covering both sparse and dense reward scenarios (Fig A.2, Table A.2 A.3) [link]](https://anonymous.4open.science/api/repo/ano/file/2.pdf). We compared DuRND with baselines suited for dense-reward tasks.
>
> The results show that DuRND performs robustly in continuous-control and dense-reward tasks, confirming its generality beyond the discrete-control and sparse-reward settings. The new experiments will be included in the revised paper.
>
> ---
>
> Regarding the suggestion on Figure 6, we have added [RND and ReLara in the ablation plot (Fig A.8) for easier comparison [link]](https://anonymous.4open.science/api/repo/ano/file/8.pdf), and it will be included in the revised paper.
>
> ---
>
> Regarding the question of why DuRND with only novelty reward (DuRND-only-nov) outperforms standard RND in challenging tasks like Montezuma’s Revenge and Pitfall, we offer two potential explanations:
>
> 1. **Longer training horizon**: Both Montezuma’s Revenge and Pitfall are more challenging, thus, as shown in our plots (x-axes), they were trained for 10× the duration of other tasks. The increased training time, when combined with DuRND’s three-level novelty estimation and its capacity for sustained exploration, allows the benefits of DuRND to become more pronounced relative to RND.
> 2. **Higher policy entropy and exploration diversity**: To further investigate this performance gap and provide direct empirical evidence, we analyzed the [policy entropy over the training process (Fig A.7) [link]](https://anonymous.4open.science/api/repo/ano/file/7.pdf). Our results show that DuRND-only-nov consistently maintains higher entropy than RND, with this effect being especially pronounced in Montezuma’s Revenge and Pitfall, where the gap is much larger than in other tasks. This shows that DuRND encourages more diverse action selection, thus broader exploration, which can be a direct empirical support to explain the experimental observation.
>
> ---
>
> Once again, we appreciate your valuable comments and hope our responses addressed your concerns.

---

### Official Review · Reviewer_krMZ · 2025-03-13

**Overall Recommendation:** 2

**Summary:**

The authors propose Dual Random Networks Distillation (DuRND), a reward shaping framework for sparse-reward reinforcement learning. DuRND consists of two random networks as its primary components, which simultaneously generate complementary rewards: one encouraging novelty-driven exploration, and the other measuring contributions toward task completion. Empirical results demonstrate that DuRND achieves superior performance compared to baseline algorithms in environments with sparse rewards.

**Claims And Evidence:**

Terms such as *convergence* or *optimal* appear frequently throughout the manuscript. However, there is insufficient theoretical or empirical results supporting for these claims.

**Essential References Not Discussed:**

No.

**Experimental Designs Or Analyses:**

Clear.

**Methods And Evaluation Criteria:**

Yes.

**Other Comments Or Suggestions:**

- It would be more accurate to comment in the right column of line 44 that reward feedback is provided when the task's goal or sub-goal is achieved, rather than at the end of each episode.

- The authors state that DuRND is designed for efficient exploration and stable convergence. However, it is unclear whether the performance improvements of the proposed method result from more efficient exploration or simply from maintaining exploration over a longer period. It appears more likely to be the latter. Furthermore, based on the experimental results, the agent's performance does not seem to converge.

- In the left column, line 256, the term *improving convergence* sounds somewhat awkward. It would be better to replace it with a term like *improving convergence rate*.

- The authors state that the use of novelty and contribution rewards effectively broadens the exploration horizon during the early stages of training and reinforces meaningful hidden values in later stages. An approach called Never Give Up [2], which significantly expands the exploration horizon, could serve as a good baseline for comparison with the proposed method.

- The authors claim that the proposed method operates with minimal computational overhead. However, since it employs two random networks, it naturally requires more computation compared to the original method that use only one random network. Therefore, given that RND is included as a baseline in Table 2, the term *minimal* does not seem appropriate.

**Other Strengths And Weaknesses:**

**strengths**

- The proposed method is simple and easy to understand.
- The authors provide various experimental results, along with clear visual representations (e.g., Figure 4), making it straightforward to understand the advantages of their approach.
- The empirical results demonstrate notable performance improvements compared to the baseline algorithms.

**weaknesses**

- The theoretical or empirical results provided to support the authors' claims is somewhat insufficient (e.g., optimal, convergence, efficient).
- The proposed approach does not significantly differ from existing intrinsic reward-based exploration methods, suggesting limited novelty or contribution.

**Questions For Authors:**

- In the left column, lines 101 and 104, the authors define the reward function as a function of state $s$. Typically, in Markov decision processes, the reward function $R$ is defined as a function of both state $s$ and action $a$. Is there a particular reason for defining it in this manner?

**Relation To Broader Scientific Literature:**

The proposed method showed notable performance improvement in sparse reward settings. Improved performance in sparse-reward environments enables more efficient learning and faster convergence toward desirable behaviors. This can be particularly valuable in real-world scenarios where immediate or dense external rewards are difficult to obtain.

**Theoretical Claims:**

There are no theoretical claims in the manuscript.

---

> ### Author Rebuttal · Authors · 2025-04-01
>
> Dear reviewer,
>
> Thanks for the comments. Our responses include new experiments and detailed elaboration below.
>
> Regarding the claims of optimal, convergence and efficient, we highlight that they're grounded in comprehensive empirical evidence:
> 1. **Optimal** refers to final evaluation performance. Our experiments across 12 tasks (Fig 1, Table 1 in paper) and [6 additional continuous-control tasks with sparse and dense rewards (Fig A.2, Table A.2 A.3) [link]](https://anonymous.4open.science/api/repo/ano/file/2.pdf), convered 24 tasks, 5 domains. DuRND achieves highest returns.
> 2. **Convergence** is evidenced by training curves (Fig 1, A.2, A.3), where DuRND stabilizes in most tasks, we clarify that we refer to empirical convergence.
> 3. **Efficiency** (a) sample efficiency: DuRND reaches higher returns with fewer steps (Fig 1, A.2); (b) computational efficiency: Table 2 shows DuRND has lower overhead than other RS methods, except its backbone RND/PPO. We agree "minimal" may be misleading and revise it as "lower overhead"
>
> ---
> Regarding the novelty and contribution over existing methods, we highlight that DuRND introduces a novel $R^{con}$ to complement pure novelty-based exploration. Importantly, $R^{con}$ is computed by lightweight RN modules, and the dual reward shaping achieves effective exploration-exploitation balance. To our knowledge, this synergy between two reward types has not been explored in prior work.
>
> ---
> Regarding the reason for DuRND's performance gain, it stems from **task-aligned exploitation via $R^{con}$ and more effective exploration, rather than merely extending exploration phase**:
>
> 1. **More effective (not naïvely longer) exploration.** First, from the entropy perspective, we show [additional results of DuRND and RND's entropy (Fig A.5) [link]](https://anonymous.4open.science/api/repo/ano/file/5.pdf), which reveals: (a) Entropy in both methods drops in similar period (~30% of training) with similar rates, indicating DuRND doesn't extend exploration phase. (b) in early stage, DuRND maintains higher entropy, especially in complex tasks (MZ’s Revenge), indicating higher action diversity and broader exploration. This arises from the dual-RN design: some states remain novel in one RN, enabling continued exploration when necessary. Second, from the state visitation perspective in Fig 4, exploration ends even earlier for DuRND, skewing to right side in 25-50k steps, while RND continuously visits the left side up till 50-75k steps. Last, $R^{nov}$ decay periods vary across tasks in Fig 5 (MZ’s Revenge vs Freeway), indicating an adaptive rather than a naïve prolonging of exploration.
> 2. **$R^{con}$ is crucial** as it bridges exploration and exploitation. As novelty doesn't imply usefulness, $R^{con}$ considers state's goal-reaching potential, prioritizing states that are both exploitable and novel over novel-only states. This is shown in [new results in the toy task (Fig A.6) [link]](https://anonymous.4open.science/api/repo/ano/file/6.pdf). We tracked $R^{nov}$ and $R^{con}$ every 20k steps. In later stages, DuRND and RND show high novelty for left and right ends due to less visits, but only the right end is goal, so $R^{nov}$ in RND leading to explore left is unreasonable. In contrast, DuRND’s $R^{con}$ assigns higher rewards to right side, hence $R^{nov}+R^{con}$ is more reasonable. This shows **novelty alone is insufficient, $R^{con}$ is essential**. Also, Fig 5 fruther shows $R^{nov}$ drops early (~first 20% of training), and $R^{con}$ dominates large later portions.
>
> ---
> Regarding Never Give Up (NGU), we [add it as a baseline (also DEIR suggested by reviewer XS8c) (Fig A.1, Table A.1) [link]](https://anonymous.4open.science/api/repo/ano/file/1.pdf). DuRND outperforms NGU for two main reasons:
> 1. Though NGU considers short and long-term novelty, it considers ONLY novelty, while $R^{con}$ in DuRND assesses goal-reaching value, prioritizing novel **and useful** states, rather than blindly exploring.
> 2. DuRND's exploration is also effective. Early training mainly updates negative RN; later the positive RN kicks in as positive states are collected, the pre-updated negative RN and newly updated positive RN are jointly considered, encouraging further exploration for necessary states.
>
> ---
> Regarding R(s) vs R(s,a), consider two common scenarios:
>
> 1. Image obs and discrete actions (Atari, VizDoom, 3DMaze in our paper), actions are integer IDs, it's uncommon to input IDs together with image states into networks, as they provide limited information, thus most works use R(s), e.g., DQN, RND.
> 2. Continuous-control (MuJoCo, robotics), both states and actions are vectors, it's natural to concatenate s-a as joint input to networks, well-suited for R(s,a). Our new experiments in Fig A.2 exactly used R(s,a). Another practice is ReLara.
>
> ---
> For terminology "rewards for both final and sub-goals", and "convergence rate", we clarified in paper.
>
> Thanks again and hope our response has addressed your concerns.

---

### Official Review · Reviewer_XS8c · 2025-03-13

**Overall Recommendation:** 4

**Summary:**

The paper proposes Dual Random Networks Distillation (DuRND), a novel reward shaping framework designed for efficient exploration and stable (extrinsic reward) convergence in sparse-reward reinforcement learning tasks. DuRND utilizes two lightweight random network modules, namely positive and negative Random Networks (RN), to simultaneously compute a novelty reward for exploration and a contribution inclined towards exploitation. The novelty reward encourages exploration of less-visited states, while the contribution reward assesses states based on their likelihood of yielding higher environmental rewards. They provide both performance and qualitative evaluations across 12 tasks, including high-dimensional tasks with challenging sparse rewards (e.g Atari, VizDoom, and MiniWorld). DuRND demonstrates superior performance and efficiency relative to several existing benchmarks (compared to 7 other algorithms, 3 with exploration bonus and 2 hidden value reward shapping).

**Claims And Evidence:**

The paper's claims regarding improved exploration efficiency, stable convergence, and minimal computational overhead seem to be well supported by experimental evidence across various tasks.

However, I consider as important caveat the lack of analysis concerning the sensitivity of the reward coefficients (coefficients for all types of reward intrinsic, extrinsic, R^con, R^nov, in both their setups and baselines like RND). There is a small possibility that Rcon may merely amplify extrinsic rewards, a behavior that could be simply replicated by scaling rewards. Additionally, the comparison to existing intrinsic reward (IR) literature, such as "Never Give Up" (Badia et al., 2020), could be strengthened to clarify how DuRND uniquely addresses the balance between novelty exploration and extrinsic rewards.

**Essential References Not Discussed:**

The paper insufficiently positions itself relative to intrinsic reward literature addressing the balancing problem between novelty and extrinsic rewards. Specifically, while it references "Never Give Up" (Badia et al., 2020), it fails to discuss how its approach differs in tackling this previously identified challenge, making its positioning unclear. NGU, just being one of the articles in the literature addressing this problem

**Experimental Designs Or Analyses:**

The experimental designs and analyses are overall sound.

However, It would be very important to have **sensitivity analyses concerning reward coefficients**. This omission is significant, because the contribution reward (Rcon) might inadvertently act as a simple scaling factor for extrinsic rewards. This effect  might be easier to be analysed in the toy example from Section 5.2.

Additionally, the authors could discuss explicitly the difference in value ranges between Rnov (unbounded) and Rcon (bounded between 0 and 1) and how these can be tackled, if necessary.

Would be good to include comparisons with newer IR mechanisms like DEIR (Wan et al., 2023) or "Never Give Up", strengthening significance of  the evaluation.

**Methods And Evaluation Criteria:**

The proposed evaluation criteria and methods are appropriate for the problem addressed, covering 12 diverse sparse-reward tasks effectively.

The evaluation could benefit from including even denser reward environments where existing intrinsic reward mechanisms are known to struggle, providing a stronger validation of DuRND's robustness.

**Other Comments Or Suggestions:**

On line 017, the reference to Sorg et al., 2010a does not appear to be related to sparse reward environments.

**Other Strengths And Weaknesses:**

The paper is clear, well-structured, demonstrating originality in combining dual random networks to balance exploration and exploitation.

The paper is interesting, by identifying the need for having a seperate, delayed reward for novelty of states that have higher chances of leading to extrinsic reward.

However, its significance could be considerably strengthened through explicit reward coefficient analyses. Despite these minor weaknesses, the work is detailed, thorough, and generally well-executed.

**Questions For Authors:**

Not at the moment.

**Relation To Broader Scientific Literature:**

The paper seem to relates its contributions adequately to the broader literature on reward shaping and intrinsic rewards.

However, I consider quite problematic a terminology issues I identified starting from the introduction section; the authors conflate reward shaping (RS) and intrinsic rewards, which are most often treaded differently in the literature and mostly represent different concepts.  They refer only to reward shaping, while cited papers from the introduction are clearly part of the intrinsic reward RL literature (e.g. the exploration bonus ones). Additionally, terms like "hidden state value approaches" are unconventional. Clarifying or explicitly stating the reason for introducing new terminologies would significantly enhance RL literature alignment.

**Theoretical Claims:**

No theoretical proofs seem to be presented.

---

> ### Author Rebuttal · Authors · 2025-04-01
>
> Dear reviewer,
>
> Thanks for the comments and below we provide detailed responses.
>
> Regarding the coefficient sensitivity and the nature of contribution reward:
>
> 1. **Coefficient sensitivity**: we conducted [additional experiments to evaluate the reward coefficients in Atari games (Fig A.3) [link]](https://anonymous.4open.science/api/repo/ano/file/3.pdf), varying the weights of the two rewards. Results show DuRND is robust to the coefficients, and its performance remains stable under different settings. These experiments will be included in the revised version.
> 2. **Normalization and balance with extrinsic rewards**: First, between extrinsic and intrinsic rewards, we follow RND and ReLara, which states that setting a shaped-to-environmental reward ratio of 2:1 leads to robust and stable learning, which is also a common practice in RS research. Second, for $R^{nov}$ and $R^{con}$, to control their magnitudes and ensure scale consistency, we applied a normalization mechanism described in **Appendix B.1** (will move to main text, also per suggestions from Reviewer GFgX). Specifically, based on the fact that the RN prediction errors are minimized thus will gradually decrease in training, we record the average error during a burn-in phase as an estimate for the upper bound, and normalize both $e_P$ and $e_N$ by twice this value. This bounds the $R^{nov}$ lies within $[0, 1]$ empirically. And as $R^{con}$ is a ratio, it is also bounded in $[0, 1]$. This design guarantees stable and comparable reward magnitudes across different tasks.
> 3. **$R^{con}$ is not a simple scaling of env rewards**, rather it densifies the sparse rewards into task-relevant dense rewards by assessing the goal-reaching likelihood. As suggested, we analyzed this effect by [new results in the toy task (Fig A.6) [link]](https://anonymous.4open.science/api/repo/ano/file/6.pdf), tracking $R^{nov}$ and $R^{con}$ for every 20k steps. We observe that both left and right ends got high $R^{nov}$ due to less frequent visits, but only the right side got $R^{con}$, and it increased along nearer to the goal, effectively densifying the sparse signal in a task-relevant value. This also evidences that **novelty alone is insufficient, and $R^{con}$ is essential for goal-directed exploitation**.
>
> ---
>
> Regarding comparison with NGU (Badia et al., 2020) and DEIR (Wan et al., 2023), we added [new experiments including both as baselines [link]](https://anonymous.4open.science/api/repo/ano/file/1.pdf).
>
> NGU uses short-term and long-term novelty to guide intra-episode and entire-training exploration. DEIR considers novelty from both environmental transitions and stochastics, and agent's own behavior. Both are representative intrinsic motivation methods.
>
> In the new experiments, DuRND outperforms both NGU and DEIR, and the key reason is DuRND explicitly used contribution reward to evaluate each state's latent value, allowing it to prioritize states that are more likely to succeed task, and avoid excessive attention to novel but less meaningful ones. This effect is also evidenced in the aforementioned results in toy task (Fig A.6), where both left and right-end states receive high $R^{nov}$ but only the right side get $R^{con}$, indicating $R^{nov} + R^{con}$ is more reasonable and effective than only $R^{nov}$.
>
> The experiments will be added to paper, both NGU and DEIR papers will be cited properly.
>
> ---
>
> Regarding dense-reward tasks, though DuRND targets sparse reward, it can be naturally extended to dense-reward settings. We adopt a simple yet effective threshold-based strategy: states with $R^{env}$ exceeding a predefined threshold (0.5 in implementation) are recorded by positive RN; otherwise, negative RN. We tested this extension on [6 continuous-control tasks in MuJoCo and Robotics (sparse-reward setting is also included) Fig A.2 Table A.2, A.3 [link]](https://anonymous.4open.science/api/repo/ano/file/2.pdf), comparing with baselines suited for dense rewards. Results show DuRND remains robust, confirming its generality.
>
> ---
>
> Regarding the terminology on RS and intrinsic rewards, we acknowledge that different classification schemes exist; in our paper, we adopt a broad definition of RS, which includes any approach that integrates auxiliary reward into environmental reward, including intrinsic rewards such as exploration bonus. This follows several recent works that also categorize intrinsic exploration as part of the general RS research (e.g., Gupta et al., Unpacking Reward Shaping, NeurIPS 2022; Devidze et al., ExploRS, NeurIPS 2022, etc.). We understand that some works treat intrinsic rewards as a distinct line of research and will clarify our definition to avoid confusion. We've also revised the terms “hidden state value approaches” to better align with conventional literature.
>
> ---
>
> Regarding the reference Sorg et al., 2010, we've revised it to ensure accurate citation.
>
> Once again, thanks for your feedback and hope our responses have addressed your concerns.

---

> > ### Comment · Reviewer_XS8c · 2025-04-04
> >
> > Thank you for all the extra experiments and for taking the time to address my notes.

---

> > > ### Author Response · Authors · 2025-04-04
> > >
> > > Dear reviewer,
> > >
> > > Thanks a lot for your feedback and for raising the score from 3 to 4. We are delighted that our response addressed your concerns and sincerely appreciate your support.

---

### Decision · Program_Chairs · 2025-05-01

**Decision:**

Accept (poster)

**Comment:**

This paper proposes a method for exploration in sparse-reward environments applied to RL. It is tested on well established benchmarks and shows improvement over similar but simpler methods (eg. RND).

The reviewers generally acknowledged that the contribution is interesting and outperforms existing methods in most of the case. They appreciated the experiments and the authors provided more results during the discussion to support their claims.

However, the reviewers also pointed out a discrepancy between the strength of the claims (e.g., optimality) and the lack of theoretical results. While the experiments are interesting, they are not sufficient to substantiate such strong claims. The authors are therefore encouraged to adopt a more measured tone.